# Reproductive Intentions Affected by Perceptions of Climate Change and Attitudes toward Death

**DOI:** 10.3390/bs12100374

**Published:** 2022-09-30

**Authors:** Eleonora Bielawska-Batorowicz, Klaudia Zagaj, Karolina Kossakowska

**Affiliations:** Institute of Psychology, Faculty of Educational Sciences, University of Lodz, 90-128 Lodz, Poland

**Keywords:** climate change, reproductive intentions, death anxiety, death fascination, offspring

## Abstract

Adverse climate change poses a threat to the health of pregnant women and unborn children and has a negative impact on the quality of life. Additionally, individuals with a high awareness of the consequences of climate change may be accompanied by a fear of the inevitable end, such as a fear of death. This, in turn, may discourage planning for offspring. Thus, both the perception of climate change and fear of death can have implications for reproductive intentions. Only a few studies to date indicate that concerns about climate change, especially when combined with attitudes towards death, may influence the formation of attitudes and reproductive plans. Thus, current research is aimed at looking at reproductive intentions from the perspective of both climate change concerns and the fear of death. This study was conducted from December 2020 to February 2021. A total of 177 childless males and females (58.8%) took part in the study. The Death Anxiety and Fascination Scale (DAFS) and Climate Change Perception Questionnaire (CCPQ) were completed online. Overall, 63.8% of respondents displayed a positive reproductive intention. Multivariable logistic regression analysis found that, in addition to the young age of respondents, the likelihood of positive reproductive intentions increases with death anxiety and decreases with death fascination and with climate health concerns. The results indicate that both climate change concerns and the fear of death are relevant to reproductive plans—positive reproductive intentions increase with death anxiety and decrease with death fascination and with climate health concerns. The results fill the gap in the existing research on predictors of reproductive intentions and can be used for further scientific exploration and practical activities addressing the issues of the determinants of decisions about having children. The individual consequences of climate change are clearly taken into account in the context of offspring planning and, therefore, should be considered in the design of social and environmental actions.

## 1. Introduction

Climate change and its effect on both the global economy and individual lives is the subject of many scientific analyses and political as well as public debates [1,2,3]. Recently, it started to be discussed in the context of human reproduction. The link between the environment and human fertility is well documented in older demographical analyses [4] and in more recent studies. The evidence of the effect of adverse environmental conditions on fertility as well as on reproductive plans was found in countries such as Mexico [5], Indonesia [6], Bangladesh [7], and Zambia [8]. Adverse environmental changes are also considered as imminent health risk factors to pregnant persons, their fetuses, and reproductive health in general [9] and also as a public mental health issue [10]. Another wave of discussion links climate change to overpopulation. The philosophical views of antinatalism (the ethical view that negatively values procreation) are applied as the ground for arguments not only for the role of humans and overpopulation in climate change [11,12] but also as the justification for action [13,14] mainly targeted on rising awareness and choice enhancement related to individual reproductive decisions.

Awareness of adverse climate change is considered to result in psychological effects such as excessive worries or even depressive symptoms [15,16,17,18,19], often referred to as “solastalgia” [15]. Whether such worries are universally transformed into reproductive intentions and decisions remains open to investigation and discussion. Some researchers indicate that the majority of their respondents are concerned with climate change and link it to human causes [20]. Those who shared such views were younger, female, educated, politically liberal, and believed in their ability to influence environmental outcomes, which might suggest either a more antinatalistic attitude or an increase in environmental concerns and actions when becoming a parent. The latter was not confirmed in a study by Thomas et al. [21], who found that being a parent had not significantly increased environmental attitudes and behaviors. Thus, some contradiction with the legacy hypothesis (legacy left to offspring with respect to environmental quality) was found. Only those who already expressed high environmental concerns tended to behave in a more environmentally sensitive way after the birth of their first child [21]. A later study [19] confirmed, mostly among female and male younger respondents, the concerns related to the adverse effects climate change might bring to their existing or future children. Such concerns were expressed more often than general concerns related to the environmental effects of procreation. Thus, one could conclude that climate change, although noticed and considered, might not entirely affect individual reproductive choices. Studies that addressed precisely this issue provided more straightforward evidence. More than 30% of Australian women under 30 years of age claimed that they would reconsider having children due to an unsafe future related to climate change [22]. For Canadian students, environmental concerns and pollution-related health concerns were the best predictors of their less-positive attitudes toward having offspring [23]. A similar link between concerns related to climate change and reproductive attitudes was found in the study conducted in New Zealand and the USA [24]. Thus, there is some evidence that climate change concerns shape reproductive attitudes and intentions.

To the framework that links climate change concerns with reproductive decisions, we would like to add an additional factor—the fear of death. Environmental changes might negatively affect the quality of life of the next generations and bring them additional suffering before the inevitable end. Such perspectives might activate feelings of terror and personal fragility, namely, the fear of death. The terror management theory indicates that fear of death can be buffered by several factors including shared social norms and standards [25,26]. If, as indicated by previous studies [19,20,23], those concerned with climate change less often plan to have children and such attitudes start to be advocated for (e.g., The BirthStrike Movement) [27], then those with a stronger fear of death might be more prone to such attitudes. Thus, we hypothesize that those with a strong fear of death—their own and their loved ones—might be more reluctant to opt for procreation. Considering this line of argument, we decided to conduct this study aimed at looking at reproductive intentions from the perspective of both climate change concerns and fear of death. We hypothesized that both climate change concerns and fear of death would result in less-positive reproductive intentions and both would be equally strong predictors of reproductive intentions.

## 2. Materials and Methods

### 2.1. Study Design

This cross-sectional web-based study was conducted to determine the relationship between reproductive intentions and attitudes towards death and climate change.

### 2.2. Ethical Consideration

The research procedure was performed in accordance with the Helsinki Declaration of Human Rights [28]. As the study was of an informative, cross-sectional, purely descriptive nature with healthy adult participants who were not subjected to experimental interventions or were expected to provide any biological material, no formal ethical approval was required under the country’s legislation. Nevertheless, ethical standards of the study were maintained, and participants were informed of the purpose, risks, and benefits of the survey and were told they could withdraw from the survey at any time, for any reason. All participants provided electronic informed consent prior to completing the questionnaire form. Electronic informed consent was prepared in accordance with the British Psychological Society Ethics Guidelines for Internet-mediated Research [29].

### 2.3. Inclusion/Exclusion Criteria

The following inclusion criteria were applied: being a heterosexual man or woman, not having any children (either biological or adopted), being at least 18 years and no older than 45 years of age at the time of admission to the study, lack of past or current clinical diagnosis of any psychiatric disease including depression, and being in a formal or informal heterosexual relationship at least one year prior to participating in the study. (Although the age range from 15 to 49 years is considered in the global literature [30] as the reproductive age, in the current study, we have included persons from 18 to 45 years of age. Firstly, we wanted only adults to participate. Secondly, due to the data on the decline in female fertility with age [31] and the fact that the current study was about the intention to become a parent in a certain time frame, we decided to include participants up to 45 years of age.). The study was targeted at individuals (not pairs), so there was no contraindication for participating in the study alone, as long as the criterion for being in a relationship was fulfilled. Participants were excluded if they were single or their relationship lasted less than one year, had a child, or had any experience of previous perinatal loss, as well as if they didn’t sign an electronic informed consent. As sexual orientation may affect attitudes toward parenthood and have an influence on reproductive decisions and ways to become a parent, the present study only involved heterosexual participants.

### 2.4. Procedure and Data Collection

As the study was conducted during the COVID-19 pandemic, due to the prevailing restrictions, the current data were collected via an online survey from 1 December 2020 to 28 February 2021. Participants were recruited through an advertisement posted on social media. Women and men who responded and declared an interest in participating in the study received information about the study’s aims and procedure (i.e., they were informed that the results of the study would only be used for scientific purposes, that participation is voluntary and anonymous, and that they could withdraw at any time without any penalty). Participants’ acceptance of these study conditions and expressed informed consent allowed them to switch to the electronic version of the questionnaires. Initially, a total of 181 participants completed the questionnaires. Of these, four were rejected at the initial data analysis stage due to a failure to meet the inclusion criteria for the age limit (they were older than 45 years). When using an online survey, it is much more difficult to control double observations. To minimize the risk, two treatments were used. First, an electronic form that could only be completed once was used to access a survey from a given IP address. Second, after calculating the overall scores for both measurement tools, we manually checked the raw scores for those observations with the same numerical results to exclude those that could have originated from the same respondent. In no case were the raw scores identical, so they could all be included in the statistical analyses. Finally, the data of 177 participants (including 104 women) who fully completed the questionnaires and met the eligibility criteria were analyzed.

### 2.5. Measures

#### 2.5.1. Sociodemographic Questionnaire

Items in the sociodemographic questionnaire were selected so as to cover key issues that may affect reproductive intentions. According to a national survey [32], decisions regarding having children are mainly influenced by: age (such plans are most often declared by people aged 18 to 24, and seldom by people aged 40 or more); gender (the greatest disproportion between men and women can be found among people aged 30 or more); education (people with higher education plan to have children more often than others); financial situation, which mainly affects the propensity to have two or more children. In the sociodemographic questionnaire, in addition to age, gender, education, and financial situation, the questions were also asked about the length of the relationship, living with a partner, declared reproductive plans (i.e., how many years from now the respondent plans to have children and how many children they plan to have), as well as the opinion of whose help the respondent could count on in the case of having children (“Regardless of your current plans, if you would decide to have a child, whose help in daily care could you count on?”). Reproductive intentions were assessed by the response to the question: “Are you and your partner planning to have children?”, where the answer “yes” was then categorized as positive intention.

#### 2.5.2. Death Anxiety and Fascination Scale (DAFS)

Death Anxiety and Fascination Scale [33] was developed to assess human attitudes to death. The DAFS consists of 23 items scored on a scale ranging from 1 (strongly disagree) to 4 (strongly agree) and comprises two scales: death anxiety (DA) and death fascination (DF). Death anxiety (nine items) refers to a general fear of death, especially related to oneself. Death fascination (14 items) contains both cognitive interest in death and dying and an acceptance of committing suicide and declared death desire. A validation study performed with 725 participants revealed satisfactory internal consistency—Cronbach’s α was 0.80 for death fascination and 0.90 for death anxiety. Both scales are independent of each other, and their time stability after one month is high (respectively *r* (46) = 0.76; *p* < 0.001 for DA and for DF *r* (46) = 0.78; *p* < 0.001) [31]. In our sample, Cronbach’s α values for DA and DF scores were 0.84, and 0.92, respectively.

#### 2.5.3. Climate Change Perception Questionnaire (CCPQ)

The Climate Change Perception Questionnaire was created to provide information on how participants respond to climate change. The questionnaire items were designed on the basis of an analysis of the content of internet portals related to climate change, published interviews, and commentaries related to the possible effects of climate change. On that ground, several indicators of the perception of progressive climate change were identified. These were: seeking/avoiding information on climate change, the migration crisis, economic crashes, active/passive attitude in relation to climate change, air quality, physiological changes (sleep, immunity), as well as the risk of contracting diseases (e.g., viral infections), which served as the basis for 21 items for the first version of the questionnaire. The final version of the CCPQ was created as a result of three successive stages of work: factor analysis, reliability determination (Cronbach’s α), and a stability assessment that was carried out in a group of 90 people (52 women and 38 men). The factor analysis revealed two subscales: (1) climate preoccupation (related to information seeking, involvement in climate action, and accompanying concerns about the socio-economic consequences of climate change) and (2) climate health concerns (related to the consequences of climate change for somatic health), with satisfactory reliability: Cronbach’s α was 0.79 for the first subscale and 0.77 for the second. The stability assessment of CCPQ was tested with 10 people (5 women and 5 men). After two weeks, the time stability of CCPQ was high; for the overall result, *r* (10) = 0.90; *p* < 0.001, for scale 1 (climate preoccupation), *r* (10) = 0.95; *p* < 0.001, and for scale 2 (climate health concerns), *r* (10) = 0.66; *p* < 0.001. This indicates the high stability of all scales and sufficient psychometric properties. The final version of the tool used in this study consists of 12 items with a four-level response scale, from 1 (definitely not) to 4 (definitely yes). In addition to the results obtained for particular subscales, it is also possible to calculate the total score of the CCPQ (the scores range from 12 to 48). The higher scores indicate more concerns about climate change. In the current sample, Cronbach’s α values for the climate preoccupation subscale, climate health concerns subscale, and the total scores were 0.83, 0.78, and 0.87, respectively.

### 2.6. Data Analysis

All statistical analyses were performed using the Statistical Package for the Social Sciences (SPSS) version 25.0 for Windows. Demographic characteristics were presented as the mean (standard deviation, SD) for continuous variables and frequency counts (percentages) for categorical variables. The chi-square test was then used to estimate the significance of differences in participants’ characteristics according to reproductive intention. The Shapiro–Wilk test was used to check the normality of distributions for all analyzed variables. Cohen’s *d* was used to determine the effect size for two means. Finally, as our outcome variable (the positive reproductive intention) was dichotomic, single-factor and multi-factor logistic regression models were tested to find its predictors. All reported numbers are based on unweighted data, and percentages, standard errors, adjusted odds ratios (AORs), and 95% confidence intervals (CIs) are based on weighted data. The statistical significance level was set at *p* < 0.05.

## 3. Results

### 3.1. Study Sample Characteristic

The study group comprised 177 participants (58.8% of women) aged from 18 to 44 years old (M = 25.6; SD = 4.9), of which more than half of the sample (57.1%) were aged less than or equal to 25 years. Most of the respondents were in a relationship lasting no more than 3 years (48.6%) and lived with a partner (66.1%). The participants had mostly university degrees (51.4%) and a good or very good financial situation (83.1%). Among the respondents, 113 (63.8%) had positive reproductive intentions. Half of them plan from two to three children (52%) within the next three years (28.2%). The majority of the respondents declared that they could count on their partners (90.4%), parents (62.2%), and parents-in-law (60.5%) and that they could not count on the help of other family members (66.7%) and friends and acquaintances (76.3%) in the case of having children. The detailed sample characteristics can be found in Table 1.

### 3.2. Climate Change Perception, Death Anxiety, and Death Fascination in a Study Sample

The scores on the Climate Change Perception Questionnaire (CCPQ) and Death Anxiety and Fascination Scale (DAFS) for the total sample and positive and negative reproductive intention subgroups are given in Table 2. Regarding the distribution of the variables, only climate preoccupation was normally distributed (W_(177)_ = 0.99; *p* = 0.165). However, skewness and kurtosis were also analyzed (see: Table 2), and none of the coefficients exceeded the value of +/−1. Therefore, a parametric t-test was used to calculate the differences of means for the analyzed variables between the positive and negative reproductive intentions subgroup.

The mean of climate preoccupation scores obtained by all participants was 16.8 (SD = 4.2), which, assuming a range from 7 to 28 points, can be considered as being in the middle of the scale. A similar assumption can be made in the case of climate health concerns, with a mean of 11.6 (SD = 2.9) and a range of scores from 5 to 19. A statistically significant difference was found only in the CCPQ climate health concerns subscale (t_(175)_ = −2.363, *p* < 0.05; d = 0.38). Participants from the negative reproductive intentions group (M = 12.3; SD = 2.8) showed higher scores than those with positive reproductive intentions (M = 11.2; SD = 3.0), although the effect size was small. There were no statistically significant differences between the groups in terms of climate preoccupation (t_(175)_ = −1.968, *p* = 0.051).

The mean of the death anxiety scores obtained in the total sample was 21.6 (SD = 5.4), and the mean of death fascination was 28.6 (SD = 9.7). Both scores can be considered as being in the middle of the scale. Statistically significant differences were found in the death anxiety and death fascination scores. Participants from the positive reproductive intentions group showed a higher intensity of death anxiety (t_(179)_ = 5.460, *p* < 0.001; d = 0.87) and lower intensity of death fascination (t_(179)_ = −3.790, *p* < 0.001; d = 0.58) than those who have negative reproductive intentions (Table 2), and the effect size was strong and moderate, respectively.

### 3.3. Predictors of a Positive Reproductive Intention

Overall, 63.8% of respondents displayed a positive reproductive intention. It was most likely reported by respondents aged up to 25 years old (61.1%), who lived with a partner (66.4%), in a relationship lasting no longer than three years (51.3%), with university education (54.9%), assessing their financial situation as good or very good (86.7%), planning to have from two to three children (81.4%), within the next one to three years (44.2%). Positive reproductive intentions were also mostly observed among those who declared they could count on their partners (90.2%), parents (70.8%), and parents-in-law (67.3%), and least in those who declared they could count on other family members (38.9%) and friends and acquaintances (22.1%). Similarly to respondents with positive reproductive intentions, a majority of those who displayed negative reproductive intentions were no older than 25 years (50.0%), live with a partner (65.6%), assess their financial situation as good or very good (76.6%), declare they could count on their partners (87.5%), and could not count on other family members (76.6%) and friends and acquaintances (73.4%). However, most respondents with a negative reproductive intention were females (62.5%) and had equally level of college and university degrees (both 43.5%) (see: Table 3).

All sociodemographic variables except for child planning time and the number of children (which only applied to respondents with positive reproductive intentions) were individually entered into the logistic regression equation. Single-factor logistic regression found six variables to be significant (Table 4). Participants ≤ 25 years were four times (OR = 4.31; 95% CI: 1.01–18.35) more likely to have a positive reproductive intention than older participants. Participants who declared they could count on their parents-in-law’s support were about 2.5 times more likely to have positive reproductive intentions (OR = 2.45; 95% CI: 1.32–4.55), and other family members’ support was twice as likely (OR = 2.07; 95% CI: 1.05–4.07) than in those who could not count on their parents-in-law. The likelihood of positive reproductive intentions increases with death anxiety (OR = 1.19; 95% CI: 1.11–1.28) and decreases with death fascination (OR = 0.95; 95% CI: 0.92–0.98), and with climate health concerns (OR = 0.89; 95% CI: 0.80–0.99). Gender, living with a partner, duration of the relationship, education level, financial situation, declaration of support expected from partners, own parents, and from friends and acquaintances, and climate preoccupation were not found to be significant factors in the single logistic regression analysis.

A multi-factor logistic regression was then carried out to assess the effect of age, parents-in-law and other family members’ support, climate health concerns, and death anxiety and death fascination on the likelihood of positive reproductive intentions (Table 5). The overall model was statistically significant when compared to the null model, (χ2_(7)_ = 61.670, *p* < 0.001), explaining 40% of the variance in positive reproductive intentions (Nagelkerke R^2^), and correctly predicted 75.7% of cases. Among all variables found to be significantly associated with positive reproductive intentions in the single-factor analysis, only age (*p* < 0.05), climate health concerns scores (*p* < 0.01), death anxiety scores (*p* < 0.001), and death fascination scores (*p* < 0.05) were significant, but declarations of support expected from parents-in-law (*p* = 0.358) and other family members (*p* = 0.184) were not (Table 5).

Respondents ≤ 25 years old were nearly six times (AOR = 5.83; 95% CI: 1.11–30.71) more likely to have a positive reproductive intention than older participants. Similarly, as in the single-factor logistic regression model, the likelihood of positive reproductive intentions increases with death anxiety (AOR = 1.27; 95% CI: 1.15–1.39) and decreases with death fascination (AOR = 0.96; 95% CI: 0.92–1.00) and with climate health concerns (AOR = 0.77; 95% CI: 0.66–0.89).

## 4. Discussion

The findings of our study indicate that participants express moderate climate concerns both with respect to their preoccupation with climate change and its effect on their health. Only the second was significantly different; thus, participants with positive reproductive intentions were less concerned with the adverse effect of climate change on their health. Scores related to death anxiety and death fascination were significantly different for groups with opposite reproductive intentions, and participants with positive reproductive intention showed a higher intensity of death anxiety and lower intensity of death fascination than those with negative intentions. Thus, we found confirmation for the hypothesis that persons who want to have children express different climate change concerns and different intensities of the fear of death than those with the opposite reproductive intentions. Our results are partly in line with those studies that, like Arnocky et al.’s study [23], point to less-positive reproductive intentions in persons more concerned with climate change, or like Schneider-Mayerson and Leong’s study [19], indicate concerns with the effect of climate change on future children and with the carbon footprint of procreation. However, our participants were concerned, rather, with the effect climate change might pose on them than on their future children. Our findings indicate that those with stronger concerns about climate change’s effect on their health are less prone to opt for having children. 

Although the scores for fear of death and fascination with death were different for those with opposite reproductive intentions, the direction of such differences was rather surprising. Considering terror management theory, we assumed that an increased fear of death might be linked to negative reproductive intentions, as becoming childfree might, on the one hand, be considered a way to subscribe to standards more and more accepted within society and one’s age group, and on the other hand, as a way to avoid the suffering of the love ones caused by climate change. On the contrary, those with positive reproductive intentions scored significantly higher on death anxiety. Therefore, it is likely that positive reproductive plans might act as a remedy for the intensive fear of death, with future children acting as one’s kind of legacy and the indicator of one’s existence. Thus, those with more intense death anxiety would more often opt for solutions that might diminish their worries related to the lack of any visible signs that they ever existed. 

The characteristics of those with positive intentions are not different from the characteristics of those who indicate negative reproductive intentions. Both groups mostly included respondents aged up to 25 years, who lived with a partner in a relationship lasting no longer than three years, were in a good or very good financial situation, and who could count on support from their partners, parents, and in-laws. Those with positive reproductive intentions more often held a university degree. Thus, the characteristics of persons who do not intend to have children, presumably partly due to climate change concerns, are mostly similar to findings reported by other authors [20,34]. Their results indicated that the decision to remain childfree is rather typical for younger and better-educated individuals, which, for education, was not confirmed by our findings. The characteristics of persons with positive reproductive intentions point to features that, like available support and a good financial situation, can make parenting easier and are irrelevant to environmental attitudes. The issue of whether prospective parents consider the environment and climate change in their reproductive decision on top of other factors remains controversial. Some studies confirm that they do [19,23], while others indicate that although prospective parents might express ecological attitudes and behaviors as well as climate-related beliefs, these do not change and definitely do not intensify after the birth of their offspring [20,21,34]. Thus, it is plausible that climate change concerns might affect reproductive decisions far less than is assumed in some theoretical analyses [3,12]. It is also possible that climate concerns, although present in clinical consultations in patients’ reports [35] and climate activists’ statements, are far less acknowledged by the so-called “general public”. 

Our assumptions indicated the equally strong role of climate change concerns and fear of death as predictors of reproductive intentions. Single-factor logistic regression indicated that positive intentions were predicted by a variety of factors. When those related significantly to positive reproductive intentions were included in the multi-factor logistic regression, they jointly explained a vast amount of variance in positive intentions. However, in such an analysis, only a few of them remained significant predictors of positive reproductive intentions. These included younger age, climate health concerns, death anxiety, and death fascination. Similar to the single-factor logistic regression model, the likelihood of positive reproductive intentions increased with death anxiety and decreased with death fascination as well as with climate health concerns. Thus, both concerns—related to climate change and to death—found their place among variables predictive of positive reproductive decisions, which confirms our initial assumptions. It should be noted, though, that only one of the climate change concerns—that related to the effect of climate on one’s health—was a significant predictor. The general preoccupation with climate change did not emerge as significant in either of the logistic regression analyses. Although our participants expressed some preoccupation with climate change (as reflected in moderate scores for the preoccupation subscale of CCPQ), it was not relevant to their reproductive intentions. The literature suggests that climate change’s effect on a person’s health is important because it is related to reproductive health as well [9]. Participants might have been aware of such a link and responded accordingly in our study; therefore, the findings indicated the predictive role of health-related climate concerns, but did not indicate the importance of a general preoccupation with climate change.

Our study puts the analysis of reproductive intentions not only in the context of climate change concerns but also in the context of the fear of death. Such a perspective provides a new theoretical framework for studies of reproductive intentions and of the effects of climate change concerns. We consider that as the strength of our study. What is more, we report findings that indicate that climate change should be analyzed not only from the perspective of general concerns and preoccupation but also from a more individual approach, namely, the perceived effect of climate change on one’s health. Those two might act differently, as was clearly present in our results. The distinction between general and individual concerns related to climate change offers a new framework for analyses and, to our knowledge, is seldom used. In our study, we used a new measure—the Climate Change Perception Questionnaire—that not only presents sound psychometric properties as indicated in separate analyses, but due to its concise form and content, can be used in future studies on climate change concerns.

Although the conclusions of the study seem to be of significant importance for practice, this research is not free from limitations. The first of them is the form the research was conducted, forced by the recent epidemiological conditions. Research conducted via the internet allows, in a shorter time than traditional “paper-and-pencil” studies, to obtain answers from a significantly larger number of respondents of all ages; however, it also raises a number of doubts. Due to the lack of the possibility of “face-to-face” contact, the researcher is not able to verify the truthfulness of the information provided. Another limitation related to the online research method is sample bias. People who do not have access to the internet or are not fluent with information technology, as well as those who do not use social media, cannot take part in the study [36]. However, Callegaro et al. [37] recommend completing the questionnaires in a safe internet environment without any pressure, which may protect from social desirability bias. Another limitation is the questionnaire nature of the research—the value of the respondents ‘statements, in this case, is partly a derivative of the questions that were formulated in the survey, as well as the participants’ reflection skills and their attitudes [35]. Therefore, one should especially take into account the discrepancies related to the individual differences of the participants, i.e., the influence of social approval, attention, intelligence, or temperament. Rosenthal and Rosnow [38] indicate the occurrence of the so-called “psychological portrait” of a volunteer, created through specific characterological features distinguishing this population from the population of “non-volunteers”, which also has some consequences for the obtained results. It seems reasonable to combine questionnaire research with, for example, observing real behaviors towards climate change. Another issue that may make the conducted analyses erroneous is the small number of respondents and the associated low diversity in terms of sociodemographic characteristics, which makes the sample unrepresentative and does not allow for the generalization of the findings. In addition, the cross-sectional nature of the study precludes drawing causal conclusions. Thus, prospective longitudinal studies are recommended as the best way to explore the associations between reproductive intentions and both climate change concerns and fear of death. Finally, our study involved only heterosexual participants. A previous Polish study involving female homo-, hetero-, and bisexual participants [39] identified significant differences between those three groups of women in their attitudes concerning motherhood: becoming a mother was more highly evaluated by bisexual women than others; homosexual women are more likely to recognize the more undesirable conditions of motherhood than bi- and heterosexual ones; and the highest motivation for having children was observed among heterosexual women. Since sexual orientation may play a moderating or mediating role between the perception of climate change or attitudes toward death and reproductive intentions, further examinations should control this variable.

## 5. Conclusions

The presented study fills the gap in the existing research on predictors of reproductive intentions by adding climate change concerns and fear of death. The findings indicate that concerns related to the effects of climate change on one’s health are more important than a general preoccupation with climate change. Thus, further scientific exploration and practical activities addressing the issues of the determinants of decisions about having children should include the perception of individual consequences of climate change. Though related to reproductive intentions, our findings indicate that the individual consequences of climate change are clearly considered by people and thus should be included or even made a focal point of environmental campaigns and actions.

## Figures and Tables

**Table 1 behavsci-12-00374-t001:** Characteristics of the respondents (N = 177).

Characteristic		n	%
Age group, years			
	≤25	101	57.1
	26–35	67	37.9
	≥36	9	5.1
Gender			
	Male	73	41.2
	Female	104	58.8
Living with a partner			
	Yes	117	66.1
	No	60	33.9
Duration of the relationship, years			
	≤3	86	48.6
	3.5–6	46	26.0
	≥7	45	25.4
Education			
	Primary/vocational education	12	6.8
	College degree	74	41.8
	University degree	91	51.4
Economic situation			
	Bad or very bad	30	16.9
	Good or very good	147	83.1
Plans to have children, years			
	Not at all	64	36.2
	Within one year	11	6.2
	1–3, y	50	28.2
	4–6, y	37	20.9
	7–10, y	15	8.5
Planned number of children			
	0	64	36.2
	1	13	7.3
	2–3	92	52.0
	≥4	8	4.5
Partner’s expected support			
	Yes	160	90.4
	No	17	9.6
Parents’ expected support			
	Yes	119	67.2
	No	58	32.8
Parents-in-law’s expected support			
	Yes	107	60.5
	No	70	39.5
Other family members’ expected support			
	Yes	59	33.3
	No	118	66.7
Friends and acquaintances’ expected support			
	Yes	42	23.7
	No	135	76.3

N, full sample; n, subsample.

**Table 2 behavsci-12-00374-t002:** Descriptive statistics for climate change perception, death anxiety, and death fascination in the total sample and subsamples according to reproductive intentions.

	Total Sample (N = 177)					Positive Reproductive Intentions Group (n = 113)					Negative Reproductive Intentions Group (n = 64)				
	M	SD	Range	Sk	Kurt	M	SD	Range	Sk	Kurt	M	SD	Range	Sk	Kurt
Climate preoccupation	16.8	4.2	7–28	0.18	−0.20	16.4	4.1	7–28	0.13	0.09	17.7	4.3	9–27	0.24	−0.73
Climate health concerns	11.6	2.9	5–19	−0.03	−0.25	11.2	3.0	5–19	0.10	−0.01	12.3	2.8	7–19	−0.11	−0.59
Death anxiety	21.6	5.4	9–34	0.18	−0.45	23.2	5.1	13–34	0.21	−0.82	18.9	4.8	9–30	0.16	−0.08
Death fascination	28.8	9.7	15–56	0.64	−0.32	26.7	8.8	15–56	0.87	0.44	32.3	10.3	15–53	0.24	−0.87

Note. M—mean; SD—standard deviation; Sk—skewness; Kurt—kurtosis.

**Table 3 behavsci-12-00374-t003:** Negative and positive reproductive intention by study sample characteristics (N = 177).

Characteristic		Positive Reproductive Intention (n = 113) % (SE)	Negative Reproductive Intention (n = 64) % (SE)
Overall reproductive intention		62.4 (0.05)	36.2 (0.06)
Age group, years ^a,b^			
	≤25	61.1 (0.07)	50.0 (0.12)
	26–35	36.3 (0.09)	40.6 (0.13)
	≥36	2.7 (0.3)	9.4 (0.27)
Gender ^e,f^			
	Male	43.4 (0.07)	37.5 (0.10)
	Female	56.6 (0.06)	62.5 (0.08)
Living with a partner ^a,e^			
	Yes	66.4 (0.06)	65.6 (0.07)
	No	33.6 (0.08)	34.4 (0.10)
Duration of the relationship, years ^a,g^			
	≤3	51.3 (0.11)	43.8 (0.11)
	3.5–6	26.5 (0.15)	25.0 (0.15)
	≥7	22.1 (0.16)	31.3 (0.16)
Education ^a,b^			
	Primary/vocational education	5.3 (0.24)	9.4 (0.27)
	College degree	39.8 (0.09)	45.3 (0.12)
	University degree	54.9 (0.08)	45.3 (0.12)
Economic situation ^b,c^			
	Bad or very bad	13.3 (0.09)	23.4 (0.11)
	Good or very good	86.7 (0.03)	76.6 (0.06)
Plans to have children, years ^a^			
	Not at all	NA	100 (NA)
	Within one year	9.7 (0.26)	NA
	1–3, y	44.2 (0.12)	NA
	4–6, y	32.7 (0.14)	NA
	7–10, y	13.3 (0.22)	NA
Planned number of children ^a^			
	0	NA	100 (NA)
	1	11.5 (0.12)	NA
	2–3	81.4 (0.04)	NA
	≥4	7.1 (0.15)	NA
Partner’s expected support ^a,b^			
	Yes	92.0 (0.03)	87.5 (0.04)
	No	8.0 (0.09)	12.5 (0.12)
Parents’ expected support ^a,g^			
	Yes	70.8 (0.05)	60.9 (0.08)
	No	29.2 (0.08)	39.1 (0.10)
Parents-in-law’s expected support ^a,g^			
	Yes	67.3 (0.05)	48.4 (0.09)
	No	32.7 (0.08)	51.6 (0.09)
Other family members’ expected support ^b,d^			
	Yes	38.9 (0.07)	23.4 (0.11)
	No	61.1 (0.06)	76.6 (0.06)
Friends and acquaintances expected support ^a,b^			
	Yes	22.1 (0.08)	26.6 (0.11)
	No	77.9 (0.04)	73.4 (0.07)

N, full sample; n, subsample; SE, standard error; ^a^ Differences in positive reproductive intention significant at *p* < 0.001 level (chi-square test); ^b^ Differences in negative reproductive intention significant at *p* < 0.001 level (chi-square test); ^c^ Differences in positive reproductive intention significant at *p* < 0.01 level (chi-square test); ^d^ Differences in positive reproductive intention significant at *p* < 0.05 level (chi-square test); ^e^ Differences in negative reproductive intention significant at *p* < 0.05 level (chi-square test); ^f^ Differences in positive reproductive intention non-significant (chi-square test); ^g^ Differences in negative reproductive intention non-significant (chi-square test).

**Table 4 behavsci-12-00374-t004:** Single-factor logistic regression model predicting positive reproductive intentions (N = 177).

Characteristic		OR	*p*	95% CI for OR	
				LL	UP
Age group, years					
	≤25	4.312	0.048	1.014	18.346
	26–35	3.154	0.126	0.725	13.723
	≥36	1			
Gender					
	Male	1.163	0.629	0.631	2.144
	Female	1			
Living with a partner					
	Yes	0.860	0.944	0.497	1.791
	No	1			
Duration of the relationship, years					
	≤3	1.840	0.097	0.895	3.783
	3.5–6	1.725	0.198	0.752	3.956
	>7	1			
Education					
	Primary/vocational education	0.415	0.142	0.128	1.342
	College degree	0.702	0.273	0.373	1.321
	University degree	1			
Economic situation					
	Bad or very bad	0.541	0.127	0.631	2.144
	Good or very good	1			
Partner’s expected support					
	Yes	1.541	0.399	0.565	4.205
	No	1			
Parents’ expected support					
	Yes	1.596	0.147	0.848	3.006
	No	1			
Parents-in-law’s expected support					
	Yes	2.452	0.004	1.321	4.549
	No	1			
Other family members’ expected support					
	Yes	2.072	0.035	1.054	4.074
	No	1			
Friends and acquaintances’ expected support					
	Yes	0.852	0.657	0.421	1.727
	No	1			
Climate preoccupation		0.934	0.066	0.868	1.004
Climate health concerns		0.890	0.030	0.802	0.989
Death anxiety		1.189	<0.001	1.107	1.276
Death fascination		0.949	<0.01	0.918	0.980

Note. OR, odds ratio; CI, confidence interval.

**Table 5 behavsci-12-00374-t005:** Multi-factor logistic regression model predicting positive reproductive intention (N = 177).

Characteristic		AOR	*p*	95% CI for AOR	
				LL	UP
Age group, years					
	≤25	5.830	0.038	1.107	30.705
	26–35	3.221	0.169	0.609	17.041
	≥36	1			
Parents-in-law’s expected support					
	Yes	1.472	0.358	0.645	3.361
	No	1			
Other family members’ expected support					
	Yes	1.815	0.184	0.754	4.372
	No	1			
Climate health concerns		0.767	0.001	0.661	0.893
Death anxiety		1.267	<0.001	1.151	1.390
Death fascination		0.954	0.033	0.916	0.999

Note. AOR, adjusted odds ratio; CI, confidence interval.

## Data Availability

The datasets used and analyzed during the current study are available in an OSF data repository at https://osf.io/xakpc/?view_only=5727e1c639374dcbad5aa5085fcffba5 (accessed on 29 August 2022).

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
