# Peer review of "Reproductive Intentions Affected by Perceptions of Climate Change and Attitudes toward Death"

_behavsci, 2022, doi:10.3390/bs12100374_

Round 1

Reviewer 1 Report

General comment:

Good theoretical foundation. Very well written.

This reviewer agrees with the authors that this study has a strong point which is to demonstrate the importance of the fear of death (because of the adverse effects of climate change on individual health) and its relationship with reproductive intentions. This finding has clinical and social relevance, but some aspects detailed below need clarification, and in particular, it needs a cite to the country’s law that allows this type of study without a favorable report from an ethics committee.

Methodological aspects:

·        Theoretical foundation: It is well argued and written.

·        Research question-Objective: Working hypothesis and objective well stated.

·        Inclusion and exclusion criteria: Adequate

·        Data collection tool: The Death Anxiety and Fascination Scale (DAFS) questionnaire used is validated. The Climate Change Perception Questionnaire (CCPQ) has been developed and validated by the authors.

Regarding the web-based data collection, this reviewer wonders how they have done to check multiple submissions from the same person, for instance, does the platform used have a control system to avoid multiple submissions from the same IP address?

·        Data analysis: Adequate Cronbach's Alpha (Cronbach's α) results for both subscales of the "perception of climate change" measurement instrument. Adequate sample for the results obtained, although, as the authors state, it is not a representative sample and therefore cannot be generalized to the population.

·        Discussion: consistent with the frame of reference and well argued.

·        Limitations. They describe the limitations of the study well, but as pointed out above, it is not clear how they identified the possibility of obtaining multiple responses from the same subject.

·        Conclusions. Very precise and concrete.

·        Bibliography: Current and well-cited bibliography.

·      

·        Ethical considerations: The paper state that: As the study was of an informative cross-sectional purely descriptive nature, no formal ethical approval was required under the country's legislation. The authors should cite the law of their country, since in other countries, even if it is an informative cross-sectional study, a favorable report from an ethics committee is required.

Best regards

Author Response

Dear Reviewer,

We appreciate the effort put into reviewing our article. Our answers are below.

General comment:

Good theoretical foundation. Very well written.

This reviewer agrees with the authors that this study has a strong point which is to demonstrate the importance of the fear of death (because of the adverse effects of climate change on individual health) and its relationship with reproductive intentions. This finding has clinical and social relevance, but some aspects detailed below need clarification, and in particular, it needs a cite to the country’s law that allows this type of study without a favorable report from an ethics committee.

Methodological aspects:

  • Theoretical foundation: It is well argued and written.
  • Research question-Objective: Working hypothesis and objective well stated.
  • Inclusion and exclusion criteria: Adequate
  • Data collection tool: The Death Anxiety and Fascination Scale (DAFS) questionnaire used is validated. The Climate Change Perception Questionnaire (CCPQ) has been developed and validated by the authors.

Regarding the web-based data collection, this reviewer wonders how they have done to check multiple submissions from the same person, for instance, does the platform used have a control system to avoid multiple submissions from the same IP address?

ANSWER: Thank you for this important comment. We have introduced additional explanations in the manuscript:

When using an online survey, it is much more difficult to control double observations. To minimize the risk, two treatments were used. First, an electronic form that could only be completed once was used to access a survey from a given IP address. Second, after calculating the overall scores for both measurement tools, we manually checked the raw scores for those observations with the same numerical results to exclude those that could have originated from the same respondent. In no case were the raw scores identical, so they could all be included in the statistical analyses.

  • Data analysis: Adequate Cronbach's Alpha (Cronbach's α) results for both subscales of the "perception of climate change" measurement instrument. Adequate sample for the results obtained, although, as the authors state, it is not a representative sample and therefore cannot be generalized to the population.
  • Discussion: consistent with the frame of reference and well argued.
  • Limitations. They describe the limitations of the study well, but as pointed out above, it is not clear how they identified the possibility of obtaining multiple responses from the same subject.

ANSWER: Thank you for pointing to the issue of multiple responses – it was addressed above, and the text with more detailed information was added to the manuscript.

  • Conclusions. Very precise and concrete.
  • Bibliography: Current and well-cited bibliography.
  •      
  • Ethical considerations: The paper state that: As the study was of an informative cross-sectional purely descriptive nature, no formal ethical approval was required under the country's legislation. The authors should cite the law of their country, since in other countries, even if it is an informative cross-sectional study, a favorable report from an ethics committee is required.

ANSWER: Thank you for this important comment. The text was amended to clarify why formal approval was not required. The law requires (an updated version of Ustawa o zawodach lekarza i lekarza dentysty, Dz.U. poz. 1291) all medical experiments to be subjected to ethical consideration by ethical committees and approved before being conducted. As our study was not designed as a medical experiment thus, such formal approval was not required. Nevertheless, we tried to respect all ethical standards usually applied in psychological investigations. We described in the text all the steps taken to ensure that ethical standards were kept.

The relevant of part of the manuscript is now as follows:

The research procedure was performed in accordance with the Helsinki Declaration of Human Rights [28]. As the study was of an informative cross-sectional, purely descriptive nature, with healthy adult participants, who were not subjected to experimental interventions or were expected to provide any biological material, no formal ethical approval was required under the country’s legislation. Nevertheless, ethical standards of the study were maintained and participants were informed of the purpose, risks, and benefits of the survey and were told they could withdraw from the survey at any time, for any reason. All participants provided electronic informed consent prior to completing the questionnaire form. Electronic informed consent was prepared in accordance with the British Psychological Society Ethics Guidelines for Internet-mediated Research [29].

Thank you.

Reviewer 2 Report

Dear Authors, 

1. Anti-natalistic ideas and choices are not new. People were making the choice to not have children in the 70s because the cold war was creating fear and belief that a third world war was inevitable.  For those affected by such thought processes it was logical that the presence of nuclear arms would result in the annihilation of Europe and other areas of the world. However, the concepts and terminology will not be familiar for some readers today. It will increase the value and potential for citation of your work if the concepts and terms are explained.

Abstract

Lines 19 - 22, "The results indicate that both climate change concerns and the fear of death are relevant to reproductive plans. They fill the gap in the existing research on predictors of reproductive intentions and can be used for further scientific exploration and practical activities addressing the issues of determinants of decisions about having children." 

The abstract does not make a clear case for the need of this research.  The two factors studied are not the most obvious determinants of the decision to have children, so this has to be explained for the reader. 

2. The abstract should also state the direction in which each of the two outcome measures affects the choice of having children.  For example, the discussion clearly states that: "Although our participants expressed some preoccupation with climate change (as reflected in moderate scores for the preoccupation subscale of CCPQ), it was not relevant to their reproductive intentions."  This message along with the other findings should be clear in the abstract.

3. The abstract should state how this new information can be used to the benefit of humans.  

4. Explain why participants had to be heterosexual. 

5. Correct the spelling of Good in table 1., table 3. and table 4. under Economic situation.

6. Review the arguments in the discussion.  There are some sentences that need clarification, for example lines 383, 384, "The climate effect on a person’s health is important because it is related to reproductive health as well [9]."  You have just stated that for your participants "...climate change......it was not relevant to their reproductive intentions."   You then follow with "Participants might have been aware of such a link.....therefore findings indicated the predictive role of health-related concerns and did not indicate the importance of preoccupation with climate change." 

7. Re-write the conclusion. The second sentence mentions "they". This should refer to people mentioned in the immediately preceding sentence, but the first sentence does not discuss people at all, so again there is some confusion. "The presented study fill the gap in the existing research on predictors of reproductive intentions. However, they clearly proofed that further scientific exploration and practical activities addressing the issues of determinants of decisions about having children should include perception on individual consequences of climate change."

The conclusion should provide a clear summary of the findings, and state how and who this new information can be of benefit to.  

Author Response

Dear Reviewer,

We appreciate the effort put into reviewing our article. Our answers are below.

Dear Authors, 

  1. Anti-natalistic ideas and choices are not new. People were making the choice to not have children in the 70s because the cold war was creating fear and belief that a third world war was inevitable.  For those affected by such thought processes it was logical that the presence of nuclear arms would result in the annihilation of Europe and other areas of the world. However, the concepts and terminology will not be familiar for some readers today. It will increase the value and potential for citation of your work if the concepts and terms are explained.

Thank you very much for pointing to the long history of antinatalistic ideas. The full presentation of this issue with relevant historical and philosophical background would extend the manuscript and was not among our aims. Thus we decided not to extend the introduction in that respect, but add only short explanation for the term antinatalism in brackets, as in the case of the legacy hypothesis.

Thus the relevant sentence will be as follows:

The philosophical views of antinatalism (the ethical view that negatively values procreation) are applied as the ground for arguments not only for the role of humans and overpopulation in climate change [11, 12] but also as the justification for action [13, 14] mainly targeted on rising awareness and choice enhancement related to individual reproductive decisions. 

Abstract

Lines 19 - 22, "The results indicate that both climate change concerns and the fear of death are relevant to reproductive plans. They fill the gap in the existing research on predictors of reproductive intentions and can be used for further scientific exploration and practical activities addressing the issues of determinants of decisions about having children." 

The abstract does not make a clear case for the need of this research.  The two factors studied are not the most obvious determinants of the decision to have children, so this has to be explained for the reader. 

  1. The abstract should also state the direction in which each of the two outcome measures affects the choice of having children.  For example, the discussion clearly states that: "Although our participants expressed some preoccupation with climate change (as reflected in moderate scores for the preoccupation subscale of CCPQ), it was not relevant to their reproductive intentions."  This message along with the other findings should be clear in the abstract.
  2. The abstract should state how this new information can be used to the benefit of humans.  

ANSWER: Thank you. The proposed changes have been included and introduced in the Abstract:

Abstract: Adverse climate change poses a threat to the health of pregnant women and unborn children and has a negative impact on the quality of life. Additionally, individuals with a high awareness of the consequences of climate change may be accompanied by a fear of the inevitable end, such as a fear of death. This, in turn, may discourage planning offspring. Thus, both the perception of climate change and fear of death can have implications for reproductive intentions. Only a few studies to date indicate that concerns about climate change, especially when combined with attitudes towards death, may influence the formation of attitudes and reproductive plans. Thus, current research is aimed at looking at reproductive intentions from the perspective of both climate change concerns and the fear of death. The study was conducted from December 2020 to February 2021. A total of 177 childless males and females (58.8%) took part in the study. The Death Anxiety and Fascination Scale (DAFS) and Climate Change Perception Questionnaire (CCPQ) were completed online. Overall, 63.8% of respondents displayed a positive reproductive intention. Multivariable logistic regression analysis found that, in addition to the young age of respondents, the likelihood of positive reproductive intentions increases with death anxiety and decreases with death fascination and with climate health concerns. The results indicate that both climate change concerns and the fear of death are relevant to reproductive plans – positive reproductive intentions increase with death anxiety and decrease with death fascination and with climate health concerns.  The results fill the gap in the existing research on predictors of reproductive intentions and can be used for further scientific exploration and practical activities addressing the issues of determinants of decisions about having children. The individual consequences of climate change are clearly taken into account in the context of offspring planning and, therefore, should be considered in the design of social and environmental actions.

4. Explain why participants had to be heterosexual. 

ANSWER: Thank you for your comment. We provided an additional explanation in the Inclusion/exclusion criteria section:

As sexual orientation may affect attitudes toward parenthood and have an influence on fertility decisions, reproductive decisions and ways to become a parent, the present study only involved heterosexual participants.

We also addressed this issue in the Limitation section:

Finally, our study involved only heterosexual participants. A previous Polish study involving female homo-, hetero-, and bisexual participants [38] identified significant differences between those three groups of women in their attitudes concerning motherhood: becoming a mother was more highly evaluated by bisexual women than others; homosexual women are more likely to recognize the more undesirable conditions of motherhood than bi- and heterosexual ones, and the highest motivation for having children has been observed among heterosexual women. Since sexual orientation may play a moderating or mediating role between the perception of climate change or attitude to death and reproductive intentions, further examination should control this variable.

5. Correct the spelling of Good in table 1., table 3. and table 4. under Economic situation.

ANSWER: Thank you. We corrected the spelling of “Good”

6. Review the arguments in the discussion.  There are some sentences that need clarification, for example lines 383, 384, "The climate effect on a person’s health is important because it is related to reproductive health as well [9]."  You have just stated that for your participants "...climate change......it was not relevant to their reproductive intentions."   You then follow with "Participants might have been aware of such a link.....therefore findings indicated the predictive role of health-related concerns and did not indicate the importance of preoccupation with climate change." 

ANSWER: Thank you for your remarks. We have reviewed the discussion section and introduced some corrections. We hope that now our arguments are more clear and more precise. In the amended part of the text, we added words and expressions that – in our view – clarify our way of thinking.

The changed part of the text is as follows:

It should be noted, though, that only one of the climate change concerns – this related to the effect of climate on one’s health – was the significant predictor. The general preoccupation with climate change did not emerge significant in either of the logistic regression analyses. Although our participants expressed some preoccupation with climate change (as reflected in moderate scores for the preoccupation subscale of CCPQ), it was not relevant to their reproductive intentions. The literature suggests that climate effect on a person’s health is important because it is related to reproductive health as well [9]. Participants might have been aware of such a link and responded accordingly in our study, therefore findings indicated the predictive role of health-related climate concerns but did not indicate the importance of general preoccupation with climate change.

7. Re-write the conclusion. The second sentence mentions "they". This should refer to people mentioned in the immediately preceding sentence, but the first sentence does not discuss people at all, so again there is some confusion. "The presented study fill the gap in the existing research on predictors of reproductive intentions. However, they clearly proofed that further scientific exploration and practical activities addressing the issues of determinants of decisions about having children should include perception on individual consequences of climate change."

The conclusion should provide a clear summary of the findings, and state how and who this new information can be of benefit to.  

ANSWER: The conclusions were reviewed and corrected according to suggestions. This part of the text is now as follows:

The presented study fills the gap in the existing research on predictors of reproductive intentions by adding climate change concerns and fear of death. The findings indicate that concerns related to the effects of climate change on one’s health are more important than general preoccupation with climate change. Thus further scientific exploration and practical activities addressing the issues of determinants of decisions about having children should include the perception of individual consequences of climate change. Though related to reproductive intentions, our findings indicate that individual consequences of climate change are clearly considered by people and thus should be included or even made a focal point of environmental campaigns and actions. 

Round 2

Reviewer 1 Report

The authors answer all the suggestions made by this reviewer.

They have improved the manuscript in general, describing some aspects in more detail. They have increased the quality of the discussion and conclusions, which work in more depth.

This reviewer already realized that the study subjects are healthy adults, that no intervention was made on them, no biological material was collected, and that they were informed of their right to withdraw at any time and for any reason. All these details were clearly deduced in the previous exposition. But these considerations are not enough to avoid passing an Ethics Committee, even if it was not a formal Ethics Committee.

It was requested to insert the citation of the state law stating that it is not necessary to go through an ethics committee in their country, and they mention the Ustawa o zawodach lekarza i lekarza dentysty, Dz.U. poz. 1291, which this reviewer has consulted without finding specific mention that studies with people, even if not experimental, should not go through an Ethics Committee. Perhaps I missed this specific issue when reading it?

Although the study subjects have given their written consent voluntarily, it would have been highly advisable to have the opinion of experts in ethics. I believe the authors should have gone through an Ethics Committee even if it was not a formal committee (it could have been, for instance, the Ethics Committee of a University).

Reviewer 2 Report

Very best wishes